# DNA and RNA Alterations Associated with Colorectal Peritoneal Metastases: A Systematic Review

**DOI:** 10.3390/cancers15020549

**Published:** 2023-01-16

**Authors:** Danique J. I. Heuvelings, Anne G. W. E. Wintjens, Julien Luyten, Guus E. W. A. Wilmink, Laura Moonen, Ernst-Jan M. Speel, Ignace H. J. T. de Hingh, Nicole D. Bouvy, Andrea Peeters

**Affiliations:** 1NUTRIM School of Nutrition and Translational Research in Metabolism, Maastricht University, 6229 ER Maastricht, The Netherlands; 2Department of General Surgery, Maastricht University Medical Center (MUMC+), 6202 AZ Maastricht, The Netherlands; 3Faculty of Science and Engineering, Maastricht University, 6229 ER Maastricht, The Netherlands; 4Department of Pathology, Maastricht University Medical Center (MUMC+), 6202 AZ Maastricht, The Netherlands; 5GROW–School for Oncology and Reproduction, Maastricht University, 6229 ER Maastricht, The Netherlands; 6Department of General Surgery, Catharina Ziekenhuis, 5623 EJ Eindhoven, The Netherlands; 7Department of Clinical Epidemiology and Medical Technology Assessment, Maastricht University Medical Centre (MUMC+), 6202 AZ Maastricht, The Netherlands

**Keywords:** biomarkers, colorectal cancer, genetic mutations, peritoneal metastases, systematic review

## Abstract

**Simple Summary:**

Colorectal cancer (CRC) patients with peritoneal metastases (PM) have a poor prognosis. Currently, research has been ongoing to develop new treatment options for CRC patients with PM. DNA/RNA alterations identification in the primary tumor might help identify patients who are at high risk for developing PM postoperatively. These patients could benefit from preventive or early treatment. The aim of this systematic review is to create an overview of studies which analyzed genomic DNA and RNA expression alteration correlated to PM with the goal of identifying potentially predictive biomarkers. We included 32 studies investigating primary colorectal tumors of 18,906 patients. Only *BRAF* mutations were reported as significantly associated with PM in 10 of 17 studies. As no specific biomarkers in the primary tumors of CRC patients could have been identified, further research with comprehensive genomic profiling is still desirable.

**Abstract:**

Background: As colorectal cancer (CRC) patients with peritoneal metastases (PM) have a poor prognosis, new treatment options are currently being investigated for CRC patients. Specific biomarkers in the primary tumor could serve as a prediction tool to estimate the risk of distant metastatic spread. This would help identify patients eligible for early treatment. Aim: To give an overview of previously studied DNA and RNA alterations in the primary tumor correlated to colorectal PM and investigate which gene mutations should be further studied. Methods: A systematic review of all published studies reporting genomic analyses on the primary tissue of CRC tumors in relation to PM was undertaken according to PRISMA guidelines. Results: Overall, 32 studies with 18,906 patients were included. *BRAF* mutations were analyzed in 17 articles, of which 10 found a significant association with PM. For all other reported genes, no association with PM was found. Two analyses with broader cancer panels did not reveal any new biomarkers. Conclusion: An association of specific biomarkers in the primary tumors of CRC patients with metastatic spread into peritoneum could not be proven. The role of *BRAF* mutations should be further investigated. In addition, studies searching for potential novel biomarkers are still required.

## 1. Introduction

Colorectal cancer (CRC) is the third most prevalent type of cancer worldwide and a common cause of morbidity and mortality, which is generally attributable to metastatic disease [1,2]. At initial diagnosis, almost one-fourth of patients with CRC present with synchronous metastases [2,3]. Liver metastases (LM) occur most frequently, followed by peritoneal metastases (PM) [2,4]. Colorectal PM are found in 5–15% of patients at primary diagnosis (synchronous PM) [2,4,5,6]. One can also develop PM after curative resection of the primary tumor (metachronous PM), usually within the first 3 years after the primary diagnosis [3]. Metachronous PM are reported in 4–12% of colon cancer patients and in 2–19% of rectal cancer patients [4,6]. However, the true incidence of PM might be underestimated. The preoperative diagnosis is mostly made by CT scan, but this has limited diagnostic accuracy for the assessment of the extent of PM [2,6,7].

CRC patients with PM have a poor prognosis. Currently, the only potentially life-prolonging treatment option involves surgical debulking of all visible metastases (cytoreductive surgery; CRS) followed by Hyperthermic Intraperitoneal Chemotherapy (HIPEC). Only a highly selected group of patients are eligible for this intervention. Patients with a poor physical condition and/or a too extensive metastatic disease are generally excluded and will undergo palliative systemic treatment or best supportive care only [2,8,9]. Without any treatment, the average life expectancy is 6 to 12 months after diagnosis [5,8,10].

Recently, research has been ongoing to develop new treatment options for locally advanced CRC patients [11]. Since these new treatment techniques could be invasive to a certain degree and be expensive, it would not be desirable to implement these routinely for all patients. A diagnostic tool able to identify patients who are at high risk of developing metachronous PM would allow targeted treatment in a preventive and/or curative setting [2]. According to previous research, a molecular profile of the primary tumor might help identify patients who are at high risk. It is hypothesized that specific biomarkers identified in the primary tumor can be incorporated in a prediction tool to estimate the risk of distant metastatic spread [12,13]. In patients with synchronous PM, genetic alterations could be interesting to determine prognosis or to predict response to therapy.

It is known that several pathogenic mutations occur during adenoma-to-carcinoma transformation in CRC. Important oncogenes are *adenomatous polyposis coli* (*APC*), *tumor suppressor gene TP53*, *Kirsten rat sarcoma virus* (*KRAS*), *transforming growth factor beta* (*TGF-β*), and *phosphatidylinositol-4,5-bisphosphate 3-kinase catalytic subunit alpha* (*PIK3CA*) [14,15]. Recent data suggest mutations may also affect the metastatic dissemination of tumors [16]. Different omics techniques, such as genomics (e.g., next-generation sequencing (NGS), polymerase chain reaction (PCR), pyrosequencing (PS), Sanger sequencing (SS)) and transcriptomics (e.g., NGS), could be used to elucidate DNA markers and RNA transcripts, respectively. Furthermore, individual omics techniques can be integrated into multi-omics analyses, which capture the complexity of diseases on multiple levels. As sequencing technologies have become less expensive, tumor genotyping has become standard practice for metastatic CRC (mCRC) [14,16]. As a result, clinicians now often have information on the mutational status of several oncogenes, and investigating molecular changes in primary tumors concerning metastatic potential is becoming more common [16,17]. We hypothesize that specific biomarkers, based on DNA/RNA alterations identified in the primary tumor, might characterize colorectal PM patients. Once identified, these alterations can be incorporated into a prediction tool to estimate the risk of PM development, prognosis, and be helpful in choosing the appropriate treatment options [12,13].

In this paper, the authors aim to systematically review the available literature to: (1) create an overview of previously investigated DNA and RNA alterations in the primary tumor correlated to colorectal PM and (2) investigate which gene mutations are of potential biomarker value and should be further studied. This study focuses solely on CRC (stages I–IV) and does not include other types of neoplasms.

## 2. Methods

### 2.1. Study Protocol and Registration

This systematic review was conducted and reported according to the guidelines of the “Preferred Reporting Items for Systematic Reviews and Meta-Analyses” (PRISMA) [18]. The study protocol was registered at PROSPERO (registration number CRD42021297366).

### 2.2. Search and Information Sources

A literature search was performed on the 6 January 2022 and repeated before submission on the 3rd of November 2022. PubMed, Embase, the Cochrane Library, and CINAHL Database were searched with the use of MeSH-, Emtree-, and free terms including “colorectal neoplasms”, “peritoneal neoplasms”, “mutations”, “genetic testing”, “genetic association studies”, “gene expression profiling” and “biomarkers, tumor” and additional search terms such as “colorectal”, “adenocarcinoma”, “carcinomatosis” and “predictive biomarker”. The full search strategy is displayed in Appendix B. A professional clinical librarian was involved to ensure an appropriate search strategy. Reference lists of all relevant publications were hand-searched for additional studies. This method of cross-referencing was continued until no further relevant publications were identified.

### 2.3. Selection Process

#### 2.3.1. Inclusion and Exclusion Criteria

Articles containing original data concerning genomic analyses on patients with CRC and PM were considered eligible. The primary outcome measure was specific mutations on the DNA or RNA level in the primary colorectal tumor that might be associated with PM. Studies were excluded if the tumor samples were not from primary tumor tissue origin or if the researchers only investigated metastases other than peritoneal ones. The method of genomic analysis was not a criterion for exclusion. Secondary sources such as technical descriptions, letters to the editor, conference proceedings, and commentaries were not considered. Only articles in English, Dutch, French, Italian, or German were eligible.

#### 2.3.2. Study Selection

All search results were imported into a free web tool designed for systematic reviewers (Rayyan) [19]. All duplicates were removed. The screening of studies for eligibility was performed by two reviewers (DH, JL) independently, using the predefined inclusion and exclusion criteria. First, articles were screened based on title and abstract. Disagreements between reviewers were resolved by initial discussion to create consensus. If the eligibility criteria were met after full-text screening by both reviewers, article inclusion followed. All references were stored in the Endnote Reference Management Tool.

#### 2.3.3. Data Items and Collection Process

Two reviewers (DH, JL) independently extracted data from the text, tables, and figures in a standardized, predefined datasheet. Data extraction for each article included first author, year of publication, country, study design, study period, inclusion and exclusion criteria, aim of the study, number of patients and genes, general patient information, methods of genomic analysis, methods of tissue collection and sample information, and outcome of genetic analysis. Data acquired via the outlined search strategy are summarized in tables.

#### 2.3.4. Study Risk of Bias Assessment

To assess the validity of the included studies, the bias risk was assessed independently by two reviewers (DH, JL). Since there is no standard bias assessment tool for the type of included studies, a suitable tool was designed based on the Risk of Bias using the Quality In Prognosis Studies (QUIPS) tool. All types of bias were evaluated and judged as low, moderate, or high risk.

## 3. Results

### 3.1. Study Selection

The electronic search yielded 1751 articles after removing duplicates. After abstract reading, 64 potentially eligible articles remained, based on the predefined inclusion and exclusion criteria. Full-text assessment from ten articles was not possible (e.g., language restrictions, congress submissions), whereafter 54 articles remained eligible. Reference checking resulted in one additional study, attaining 55 articles for full-text assessment. As 23 articles did not fulfill inclusion criteria, 32 studies were included for final analysis. No additional publications were identified after repeating the search before submission. The study selection process is summarized in Figure 1.

### 3.2. Study Characteristics

All 32 studies are observational cohort or case control studies published between 2008 and 2021. The number of subjects per study ranged from 15 to 5967, with a total of 18,906 patients. The main characteristics of the included studies are summarized in Table 1. In 21 studies, the tissue samples were retrieved retrospectively [20,21,22,23,24,25,26,27,28,29,30,31,32,33,34,35,36,37,38,39,40]. Two studies collected tissue samples at time of surgical resection [41,42], and in nine studies, there was no need for tissue collection because the mutation status of genes of interest had already been analyzed as part of diagnostic reasons [43,44,45,46,47,48,49,50,51]. Most tissue samples used in the studies (*n* = 24) were formalin-fixed paraffin-embedded (FFPE), while the remaining eight studies used fresh frozen tumor samples [22,31,32,34,38,40,41,42]. All characteristics of patients and tissue samples are summarized in Table 2. Only two articles reported the time of PM occurrence, i.e., metachronous or synchronous metastases [23,44]. All other studies did not specify the time of onset of PM or only included synchronous metastases. Because of the heterogeneity among the included studies in terms of the study population, genetic analyses methods, level of genetic testing, and (number of) genes, pooling in a meta-analysis was not possible.

### 3.3. Risk of Bias in Studies

The relevant categories from the QUIPS tool were used to access the risk of bias; a score per domain per study is presented in Figure 2A. We reported a high risk of bias for five studies [20,33,35,37,45] and a moderate risk of bias for all other studies. The overall lowest risk of bias was found in the statistical and outcome measurement domains, while the highest was found in the confounding domain (Figure 2B).

### 3.4. Reported Genes

Most studies focused on a selected predefined group of genes (Figure 3). Genes that were predominantly studied were *RAS* (*KRAS*/*NRAS*), *PIK3CA*, *TP53*, and *BRAF*. The remaining 13 genes (e.g., *androgen receptor* (*AR*), *ASXL Transcriptional Regulator 1* (*ASXL1*), *AT-Rich Interaction Domain 1A* (*ARID1A*), *NIMA Related Kinase 2* (*NEK2*), *MET Transcriptional Regulator MACC1* (*MACC1*), *Paired Box 5* (*PAX5*), *Ubiquitin Protein Ligase E3 Component N-Recognin 5* (*UBR5*), *Vimentin*, *Ret Proto-Oncogene* (*RET*), *Histone acetyltransferase* (*Tip60*), *PKHD1 Ciliary IPT Domain Containing Fibrocystin/Polyductin* (*PKHD1*), *Regenerating Family Member 1 Alpha* (*REG1A*), and *Kinesin Family Member 18A* (*Kif18A*)) were, except for *ARID1A*, all separately examined by individual studies (Figure 4). Three studies did a broader comprehensive genomic analysis on the tissue samples. Jacob et al. performed a PanCancer Progression Panel in 2 studies [25,26] including 770 genes, and Lee et al. used a Comprehensive Cancer Panel covering 409 genes [30]. All details about the reported genes are displayed in Table 3.

### 3.5. Genetic Analysis Methods

Primary tumor genetic analysis was performed on DNA level in 19 studies and on RNA level in seven studies (Figure 5). One study described the analysis on both levels [35]. Heublein et al. investigated MicroRNAs (miRNA) and the corresponding overexpression profiles [24]. Four articles did not specify if they performed testing on DNA or RNA level [43,46,47,52]; three of these articles did not even specify which method they used for genetic testing [43,46,52]. Nine articles reported the use of real-time polymerase chain reaction (RT-PCR) [20,21,24,31,32,35,38,42,48]. One study specified the PCR tool as quantitative methylation-specific PCR (qMSP) [41]. PS and NGS were used in one [37] and five [23,28,30,34,50] studies, respectively. Christensen et al. reported the use of both methods [44]. Two studies analyzed the samples with SS [47,49]. Jacob et al. described NanoString analysis in both their articles [25,26]. All details about the genetic analyses are displayed in Table 3.

### 3.6. DNA/RNA Alterations Outcomes and Association with PM

All details about the reported alterations are displayed in Table 3.

#### 3.6.1. *Mitogen-Activated Protein Kinase* (*MAPK*) Pathway Outcomes

*BRAF* and RAS are both involved in the *MAPK* pathway and were most commonly reported. *BRAF* mutations were analyzed in 17 articles [21,22,23,27,29,33,34,35,37,39,40,43,44,45,46,48,49]. In ten studies, it was found on a statistically significant level that *BRAF* mutant tumors were more likely to develop PM and/or that patients with PM had more often *BRAF* mutated primary tumors compared to PM-free CRC patients [22,27,33,34,39,40,45,46,48,49]. Most studies conducted the *BRAF* mutation analysis on codon 600, exon 15 (*n* = 12). Taniguchi et al. reported that the frequencies of *BRAF* mutations, in combination with *RAS* wild-type (WT) tumors, were significantly higher in CRC patients with PM [39]. Smith et al. showed a statistically significant association when *BRAF* status in unresectable CRC patients with PM was compared to other metastatic sites. This result, however, did not remain significant after a post hoc Bonferroni correction [37]. The authors also mention that *BRAF* mutations were significantly more common in patients with peritoneal-only metastases compared to patients with liver-only metastases. This, however, did not withstand a correction for multiple testing [37]. Atreya et al. and Bruzzi et al. reported no statistically significant difference in metastatic sites and *BRAF* mutation, although PM were more commonly observed in patients whose tumors harbored a *BRAF* mutation [21,43]. He et al. investigated therapy-naïve synchronous mCRC patients and found no significant differences in mutation status [23]. Shelygin et al. found no association between PM and *BRAF* status when comparing patients, with and without PM, undergoing surgery for CRC [35]. Christensen et al. looked at the probability of developing PM while having a *BRAF* mutated tumor. The hazard ratio for developing PM and having a *BRAF*-mutated tumor was statistically not significant [44]. One article did not report any data about *BRAF* mutations and its relation to PM, although they intended to investigate this [28].

RAS pathway mutation analyses were reported in 14 studies. Seven studies focused on both *KRAS* and *NRAS* genes [21,27,28,29,34,37,44], and the other seven studies only described *KRAS* variants [23,33,35,40,45,47,51]. Lan et al. reported that the proportion of PM was significantly higher in stage I–IV CRC patients whose tumors carried a RAS pathway mutation, and *KRAS*-mutated tumors had a trend toward a higher proportion of PM, which was not significant [28]. Both Zihui Yong et al. and He et al. found a significant association between *KRAS* mutant tumors and PM [23,51]. He et al. also stated that therapy-naïve synchronous PM patients tend to carry a mutant *KRAS* codon 12 [23]. One article did not report any outcomes, although they aimed to do so [28]. All other studies did not find a significant association or trend between *KRAS*/*NRAS* mutant tumors and the development of PM [21,27,33,34,35,37,40,44,45,47].

To conclude, most articles (*n* = 10/17) state that *BRAF* mutant tumors are more likely to have PM and/or mutations in *BRAF* were more common in patients with PM compared to those without. Almost all articles (*n* = 10/14) state that RAS pathway mutated tumors are not likely to have PM and were not more common in patients with PM compared to without PM.

#### 3.6.2. *PIK3CA* Outcomes

The potential association of *PIK3CA* mutations with PM was analyzed in seven studies. In five studies, the *PIK3CA* mutations were not significantly associated with PM [28,33,35,37,44]. Christensen et al. even found that *PIK3CA* mutations were associated with the absence of PM and a decreased hazard of developing PM (HR = 0.31; 95%CI = 0.11–0.86, *p* = 0.024) in mCRC patients who had received chemo- or immunotherapy treatments [44]. Two studies did not report any outcomes, although *PIK3CA* mutations were investigated [29,39].

#### 3.6.3. *TP53* Outcomes

*TP53* mutations were analyzed in four studies. Two studies showed a significant association between PM and *TP53* mutations. Lee et al. detected more *TP53* mutations in patients with small obstructive CRC with PM compared to large non-obstructive tumors without PM [30]. Sjo et al. performed a multivariate analysis in stage IV CRC patients and showed that PM was significantly associated with *TP53* mutations [36]. Lan et al. stated that stage IV CRC patients with PM had a higher frequency of *TP53* mutations, although the authors did not perform statistical analysis on this association [29]. Sayagués et al. did not find a significant association between *TP53* mutational status and PM in Caucasian patients diagnosed with CRC [34].

#### 3.6.4. Other DNA Outcomes

*AR*, *ASXL1*, *ARID1A*, *Kif18A*, *NEK2*, *MACC1*, *PAX5*, *PKHD1*, *REG1A*, *RET*, *Tip60*, and *UBR5* were mentioned as possible mutated genes associated with PM by several authors [20,29,30,31,32,38,41,50] but were, except for *ARID1A*, all investigated in only one study. NGS was performed by Yang et al. to detect RET mutations in mCRC without neoadjuvant treatment [50]. The presence of *RET* mutations was significantly associated with PM compared to WT tumors. *Tip60* regulation analysis was performed with RT-PCR in patients undergoing surgery for CRC by Sakuraba et al. [32]. The authors found that a downregulation of *Tip60* was significantly associated with PM. To conclude, all previous mentioned genes showed a significant association with PM, but all were studied by a single study only.

#### 3.6.5. RNA Outcomes

Nagahara et al. report that *Kif18A* overexpression, measured by RT-PCR, in CRC patients without neoadjuvant treatment significantly correlates with PM [31]. The expression profile of *NEK2* was analyzed by Takahashi et al. in patients with CRC who underwent surgical treatment [38], demonstrating that the high *NEK2* expression group had significantly greater peritoneal dissemination compared to the low expression group. *MACC1* expression was found to be significantly associated with PM by Shirahata et al. [42]. The expression of *REG1A* was explored in non-pretreated CRC patients by Astrosini et al. and showed a positive e correlation with the formation of PM [20]. In addition, Heublein et al. analyzed MicroRNAs (miRNAs) expression profiles and concluded that hsa-mri-31-5p seems to be overexpressed in patients with PM [24]. The authors reported a set of 31 miRNAs which were significantly upregulated in the PM group, while ten miRNAs were found to be repressed as compared to LM. Another set of two miRNAs was significantly upregulated in the PM group, while 25 were found to be repressed as compared to no metastases. Shirahata et al. discovered a trend toward preferentially developing PM in tumors with Vimentin methylation, although this was not significant [41].

#### 3.6.6. Results of Broader Panel Analyses

Lee et al. performed a broader panel analysis of which the results (*ARID1A*, *PKHD1*, *UBR5*, *PAX5*, *TP53*, *ASXL1* and *AR*) are already described in Section 3.6.4 [30]. Jacob et al. explored gene expression profiles with a broad cancer “panel” comparing four groups (without metastases, with LM, with PM, and with both LM and PM) [25]. They report that “18 genes had significantly different expression rates”, but they did not describe which genes. In another study, in which three groups were compared (without metastases, with LM, and with PM), the authors reported no significant down- or upregulation of distinct gene sets [26].

All details about the reported genes and corresponding conclusions are described in Appendix A. A conclusive summary for all genes is displayed in Figure 6.

### 3.7. MSI Status

In addition to DNA and RNA alterations, microsatellite instability (MSI) status was reported in ten articles [21,28,34,35,37,43,45,46,48,50]. Tran et al. describe the impact of *BRAF* mutations in combination with MSI status on the pattern of metastatic spread and its prognosis [48]. The authors report that patients with MSI tumors show poorer survival in mCRC, and this is due to the association with *BRAF* mutations. Yang et al. state that MSI is associated with RET mutations [50].

## 4. Discussion

This systematic review provides an overview of the results of studies which analyzed genomic DNA and RNA expression alterations correlated to PM with the goal of identifying alterations that could potentially serve as a predictive biomarker in patients with CRC. Of the 17 studies investigating *BRAF* mutations, ten studies reported a significant association with PM. Mutations in *ARID1A*, *ASXL1*, *Kif18A*, *NEK2*, *MACC1*, *PAX5*, *PKHD1*, *REG1A*, *RET*, *Tip60* and *UBR5* were also reported to be associated with PM [20,29,30,31,32,38,41,50], although these results were only described in maximum of one study. A recent analysis with a cancer panel of 770 genes from Jacob et al. did not show a significant down- or upregulation of distinct gene sets between CRC patients with PM and without distant metastases. Their sample size was, however, small (*n* = 18) [26].

### 4.1. BRAF Mutations

*BRAF* gene mutations occur in 5–15% of the mCRC cases; over 95% of these mutations consist of a substitution of valine to glutamic acid at codon 600 (V600E) [13,16,53]. *BRAF* is a serine/threonine protein kinase that plays an important role in the *MAPK* pathway. This pathway drives cell proliferation, differentiation, migration, survival, and angiogenesis, and therefore, changes in this pathway are associated with tumorigenesis [54]. *BRAF* mutations can be considered as an independent negative prognostic factor in early-stage microsatellite stable tumors and as a negative predictive factor for therapeutic approaches [54]. Due to its chemoresistance and resistance to *BRAF* inhibitor therapy, *BRAF*-mutated tumors are difficult to treat [54,55]. Therefore, trials are currently going on with dual or triple drug therapy to enhance blockade of the *MAPK* pathway. Nowadays, CRC patients without metastases are not screened for *BRAF* mutations, and further molecular examination is only conducted in metastatic disease [56]. As only 55% of the studies reported a significant association between *BRAF* mutations and PM, we cannot conclude yet that *BRAF* mutations are specific enough to identify patients with colorectal PM.

### 4.2. Other Mutations

First, RAS pathway mutations are the most commonly investigated mutations in mCRC. Different codons of both *KRAS* and *NRAS* genes were included, thereby creating a broader overview of this pathway. *KRAS* is the most commonly activated oncogene in CRC, with mutations occurring in exon 2 codon 12 and 13, exon 3 codon 59 and 61, and exon 4 codon 117 and 146 [16,57]. Approximately 30–50% of the CRC patients carry a somatic *KRAS* mutation [16]. *KRAS* mutations have been associated with lung metastases but not with PM [16]. *NRAS* is mutually exclusive with *BRAF* and *KRAS* and occurs in approximately 3% of CRC patients [16]. There has been no previously described association with PM, which is in line with the findings of this review. Second, *PIK3CA* (exon 9 and 20) gene mutations occur in 10–18% of CRC patients [53]. They commonly co-occur with *KRAS* or *BRAF* mutations. Approximately 70% of *PIK3CA* mutant patients have concurrent mutations [16,58], although they have never been described to be associated with PM. The results of our study demonstrate this as well. Third, *TP53* gene mutations are one of the most frequently described mutations as they occur in 35–75% of the colorectal PM patients [13]. Previous research shows the contradictory result of *TP53* mutations and their prognostic value in CRC patients [53]. In this review, some authors showed a significant association, while others did not reach the significance.

### 4.3. MSI Status

Of the included studies, only 10 articles reported on MSI status, all without extensive analysis. This is unfortunate, as MSI status is the only prognostic molecular marker used in deciding adjuvant therapy options [56]. MSI originates from the inactivation of mismatch repair genes by either MLH1 hypermethylation or mutation. This results in the accumulation of somatic mutations and subsequent genomic instability, which is associated with nonhereditary CRC [53]. It is well reported that MSI is a good prognostic factor for some treatments in early-stage CRC [59]. We believe it is important to always report MSI status in biomarker research to incorporate all relevant characteristics.

### 4.4. Clinical Relevancy

Clinically, the known risk factors for metachronous colorectal PM are an advanced tumor stage, right-sided tumor, infiltrative or ulcero-infiltrative tumors, history of perforation, and obstruction [3,8,60]. A randomized trial (COLOPEC-1) investigating the therapeutic effectiveness of adjuvant HIPEC to prevent PM development in high-risk CRC patients showed that this treatment strategy did not improve PM-free survival [11]. In contrast, a Spanish study by Arjona-Sánchez et al. concluded that adjuvant HIPEC therapy might be useful in patients with T4 tumors [61]. Identifying genetic alterations in high-risk metachronous PM patients may have additional benefit on improving survival by additional targeted therapies such as adjuvant HIPEC. In synchronous PM patients, the alterations provide added value to determine prognosis or to predict response to therapy. For example, RAS pathway activating mutations are negative predictive markers for the efficacy of anti-epidermal growth factor receptor (EGFR) therapies [62], while MSI tumors with *BRAF* and *PIK3CA* mutations show survival benefit [39]. For CRS and HIPEC scheduled patients, a *BRAF* mutation is a marker for poor prognosis, whereas *KRAS* tumors do not influence the outcomes [63]. The choice of cytostatic in HIPEC can be based on mutation status, or specific therapy can be developed in the case of targetable mutations.

Unfortunately, most of the studies did not clearly specify whether the authors were using tumors from synchronous or metachronous PM patients. It was therefore hard to distinguish and separate these two scenarios in the results. Future studies should clearly specify the time of metastases onset, the aim of the genetic analysis, and clinical implications.

### 4.5. Techniques

In the studies evaluated in this review, several different genetic research techniques were applied. Since most studies used targeted PCR techniques to detect specific gene mutations, the number of studies that used comprehensive genetic analyses was scarce. The development and use of NGS technologies have revolutionized the speed and throughput of DNA and RNA sequencing [64,65]. However, since the number of relevant cancer genes guiding targeted therapy in CRC is still limited and costs per sample are substantial, NGS sequencing is not yet commonly used in clinical decision making or limited to mutation hotspot target regions [66]. This has most likely influenced the research to unmap PM predictive biomarkers so far, and we believe that more comprehensive NGS analyses are needed for this purpose. When we critically look at the choice of techniques used in the included studies, we believe these were too restricted to identify DNA/RNA biomarkers in the primary tumor of CRC patients with synchronous or metachronous PM.

As mCRC is a highly complex genetic disease, an understanding of how all aspects interact is required to achieve the prediction and treatment of colorectal PM. Single target techniques, mostly used in the included articles in this paper, might be insufficient for this purpose. We believe that omics techniques (i.e., techniques that generate high-throughput data [67]) might be a promising method for new CRC biomarkers research instead of most of the methods used in this paper. The integration of multiple omics techniques, by combining genomic data with data from other modalities such as transcriptomics, epigenetics, and proteomics, to measure gene expression, gene activation, and protein levels, could be helpful to reveal this problem in further research. This integration might bring us much closer to the prediction, prevention and tailored treatment of PM in CRC [68].

### 4.6. Limitations

This is the first systematic literature review of DNA/RNA biomarkers in relation to colorectal PM to the author’s knowledge. This study has also some limitations. First, almost all included studies were retrospective with a different number of patients and different patients’ characteristics (T-stage, number of metastatic sites, treatments, etc.). Second, comparisons between the studies are limited due to heterogeneity, and a meta-analysis was therefore not possible to perform. The standardization of techniques and analysis and more insight in the individual analysis outcomes via FAIR data sharing would be helpful. Third, most studies focused on the most commonly analyzed CRC target genes, i.e., *KRAS*, *NRAS*, *BRAF*, *PIK3CA*, and *TP53* with simple sequencing methods and PCR technology. Only three studies performed a broader gene panel NGS analysis. Fourth, most of the included studies did not report if CRC patients received neoadjuvant systemic treatments and if they did, which type. Such treatments could namely affect the outcomes of the genetic analysis. Fifth, most of the studies lacked the MSI of the CRCs. Sixth, all studies showed a moderate to high risk of bias with a high risk for the confounding domain.

### 4.7. Future Perspectives

We believe the use of comprehensive genomic profiling with for example broader cancer gene panels is essential to identify new potential cancer genes for PM prediction. In addition to using an optimal technique, we recommend applying these in a homogenous patient population (e.g., strict synchronous or metachronous PM patients, tumor characteristics, etc.).

## 5. Conclusions

Increasing amount of data suggest that the presence of biomarkers in the primary tumor might have an impact on metastatic patterns. However, unfortunately, based on the given evidence, we cannot consider the genes (e.g., *BRAF*) possibly associated with PM as reliable enough to function as an individual biomarker in a clinical setting yet. Further investigation as well as more exploratory research questions leading to identify novel biomarkers, rather than performing analyses on panels consisting mostly of already established biomarkers, are still necessary. Techniques on DNA and RNA level are required to determine an association between genomic, epigenomic and transcriptomic changes and colorectal PM. Furthermore, future studies should include homogenous populations so that firm conclusions can be drawn. In that way, we might be able to identify biomarkers that can be incorporated in a prediction tool to estimate the risk of distant metastatic spread or to create targeted treatment options.

## Figures and Tables

**Figure 1 cancers-15-00549-f001:**
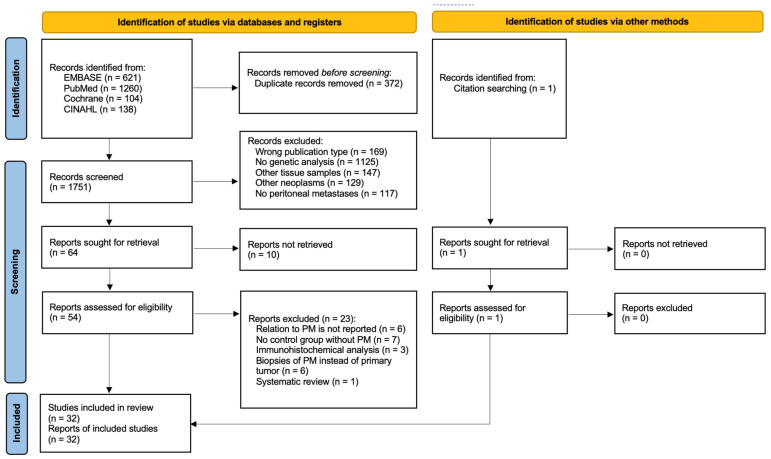
PRISMA flowchart outlining study selection strategy [18].

**Figure 2 cancers-15-00549-f002:**
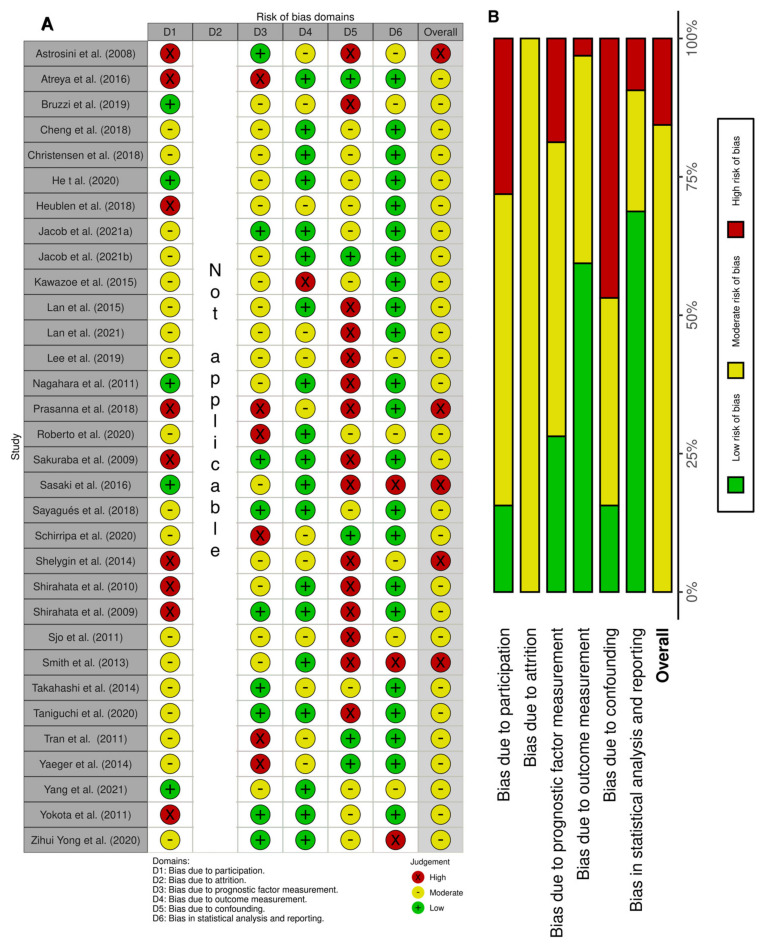
Risk of bias based on the QUIPS tool. (**A**) Summary of the domain–level judgements for each study. (**B**) Risk–of–bias judgements within each bias domain.

**Figure 3 cancers-15-00549-f003:**
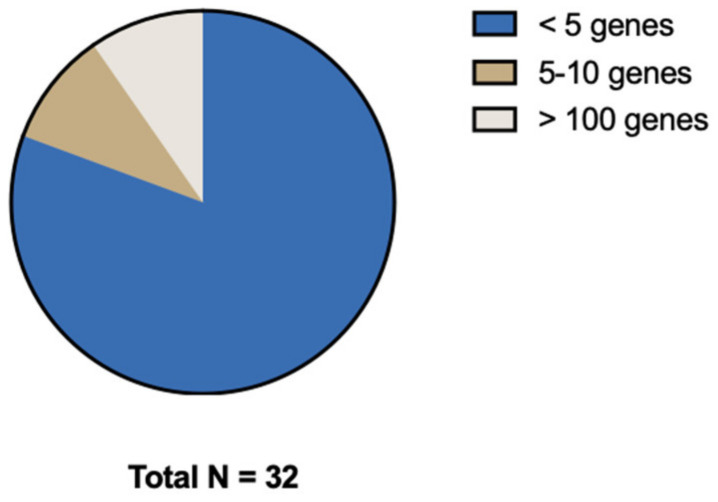
Distribution of number of genes investigated.

**Figure 4 cancers-15-00549-f004:**
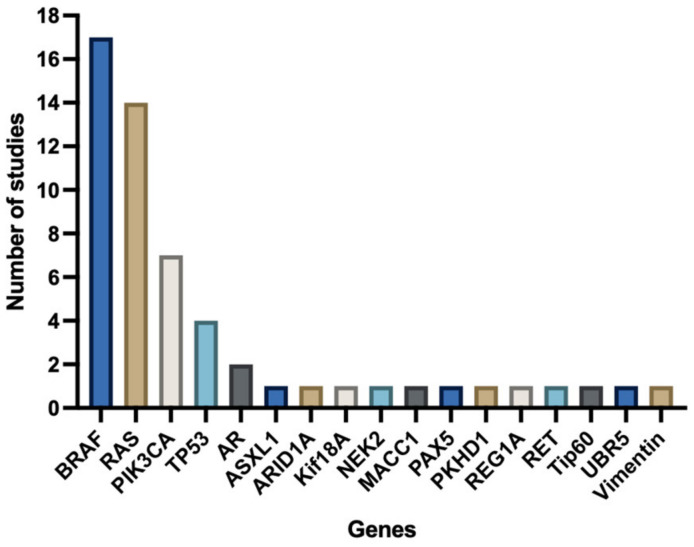
Number of studies investigating specific genes.

**Figure 5 cancers-15-00549-f005:**
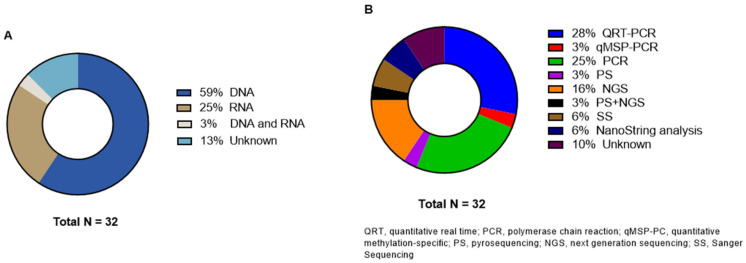
Distribution of (**A**) genetic analysis level and (**B**) different molecular techniques.

**Figure 6 cancers-15-00549-f006:**
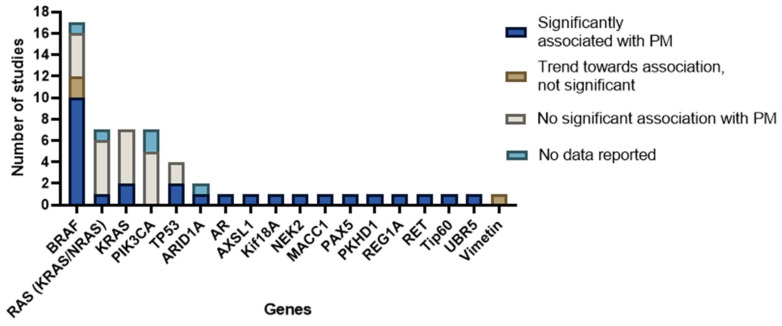
Overview of genes investigated with conclusions formulated by the authors of included studies.

**Table 1 cancers-15-00549-t001:** Characteristics of included studies.

Reference	Year	Country	Study Design	Patient Inclusion Period		Reasons for Patient Exclusion	No. of Subjects	No. of Genes Investigated	Aim of the Study
				Start	End				
Astrosini et al. [20]	2008	Germany	CH	-	-	-	63	1	To investigate if *REG1A* is upregulated in CRC patients with unfavorable clinical outcome.
Atreya et al. [43]	2016	USA	MCH	01/2013	09/2015	-	120	1	To investigate if *BRAF* V600E mutation is associated with sites and radiographic appearance of metastatic disease in patients matched by primary tumor location.
Bruzzi et al. [21]	2019	France	CH	12/2005	11/2009	-	1650	2	To assess recurrence patterns according to microsatellite instability, RAS and *BRAF*V600E status in stage III.
Cheng et al. [22]	2018	Taiwan	CH	2000	2013	History of other malignancies, inflammatory bowel disease or death within 30 days after surgery.	1969	1	To evaluate clinicopathological features, metastatic patterns, and prognostic value of CRC with the *BRAF*V600E mutation.
Christensen et al. [44]	2018	Denmark	CH	01/2005	08/2008	Presence of other active tumors and no tissue sample or medical charts available.	448	3	To investigate associations between mutations and pattern of metastases.
He et al. [23]	2020	China	CH	12/2015	02/2020	Non-metastatic synchronous CRC; neo-adjuvant therapy; location cecum, appendix or ileocecal junction; neuroendocrine components.	194	3	To investigate the connection between mutant *KRAS*, *NRAS*, and *BRAF* and clinicopathological characteristics in therapy-naïve synchronous mCRC in Chinese populations.
Heublein et al. [24]	2018	Germany	CH	1988	2012	-	23	754 miRNAs	miRNA profiling of primary CRC tissue to identify miRNAs potentially associated with defining the site of metastatic spread in CRC.
Jacob et al. [25]	2021	Germany	CC	01/2005	12/2014	Lack of any of the baseline variables or specimens, HNPCC or FAP co-malignancies.	18	770	To identify genes associated with the metastatic route in CRC.
Jacob et al. [26]	2021	Germany	CH	-	-	Missing FFPE tissue of the primary tumor, co-malignancies, Lynch-syndrome or other hereditary diseases.	18	770	To elucidate the link between immunosurveillance and organotropism of metastases in CRC by evaluating different gene signatures and pathways.
Kawazoe et al. [27]	2015	Japan	CH	01/2013	06/2014	-	264	4	To evaluate mutations in Japanese mCRC patients and assessing their corresponding effects on the efficacy of anti-EGFR therapy.
Lan et al. [28]	2015	Taiwan	CH	03/2000	01/2010	No tissue sample available in the biobank.	1492	7	To analyze mutation spectra of the *PI3K* and *RAS* pathways in CRC and the associations with sites of metastases of recurrence.
Lan et al. [29]	2021	Taiwan	CH	-	-	Patients who had stage I–III CRC, received emergent surgery, or who did not have available tumor or preoperative serum samples in the biobank.	95	10	To evaluate the concordance of mutation patterns between tumor tissue DNA and circulating cell-free DNA in stage IV CRC patients and to analyze relationship between the mutational patterns and site of metastases.
Lee et al. [30]	2019	Korea	CH	2004	2008	-	15	409	Analyzing genetic mutations which may be presage PM.
Nagahara et al. [31]	2011	Japan	CH	1993	2000	Chemotherapy or radiotherapy before surgery.	113	1	To investigate if *Kif18A* has a role in the progression of CRC.
Prasanna et al. [45]	2018	Australia	CH	01/2005	12/2015	-	5967	2	To explore the outcome of patients with mCRC based on their site of metastases at diagnosis and to explore the association between tumor characteristics and site of metastases.
Roberto et al. [46]	2020	Italy	CH	2008	2019	-	207	1	To evaluate the outcome of right CRC patients according to *BRAF* status and the treatment performed.
Sakuraba et al. [32]	2009	Japan	CH	-	-	-	38	1	To evaluate the correlation between *Tip60* expression and the clinicopathological findings.
Sasaki et al. [33]	2016	Japan	CH	02/2006	10/2011	Previous chemotherapy for advanced disease.	526	3	To compare the prognostic impact of modern chemotherapy or *anti*-*EGFR* monoclonal antibody between CRC patients with and without PM.
Sayagués et al. [34]	2018	Spain	CH	-	-	-	87	4	To investigate the frequency of mutations in primary sCRC tumors and their impact on patient progression-free survival and overall survival.
Schirripa et al. [47]	2020	Italy	CH	01/2010	12/2018	-	499	1	Description of the clinicopathologic features and prognosis of *KRAS G12C*-mutated metastatic CRC.
Shelygin et al. [35]	2014	Russia	CH	11/2012	02/2014	-	58	7	To describe the epithelial–mesenchymal transition in terms of gene expression profile and somatic changes in CRC patients with or without PM.
Shirahata et al. [42]	2010	Japan	CH	-	-	-	52	1	To examine the expression of the *MACC1* gene in primary tumors and to evaluate the correlation between the *MACC1* expression and the clinicopathological findings.
Shirahata et al. [41]	2009	Japan	CH	-	-	-	48	1	To examine the methylation status of the Vimentin gene in CRC patients and to evaluate the correlation between this status and the clinicopathological findings.
Sjo et al. [36]	2011	Norway	CH	1987	2006	Stage I–III or stage unknown, distant metastases without PM, no surgery or diverting procedures.	57	1	To evaluate the incidence of PM in CRC patients and to compare clinicopathological characteristics, survival and *TP53* mutation status in primary tumors.
Smith et al. [37]	2013	UK	CH	-	-	Patients with resectable disease.	2161	4	To study the somatic molecular profile of the epidermal growth factor receptor pathway in advanced CRC, its relationship to prognosis, the site of the primary and metastases, and response to cetuximab.
Takahashi et al. [38]	2013	Japan	CH	1992	2002	-	180	1	To investigate the clinical significance of *NEK2*/miR128 expression in CRC.
Taniguchi et al. [39]	2020	Japan	CH	08/2014	04/2016	RAS-mutant tumors or unknown RAS status.	331	2	To present institutional experience with patients with CRC who underwent clinical mutation profiling and to evaluate the differences in patient characteristics with *BRAF* mutations.
Tran et al. [48]	2011	Australia	CH	-	-	-	524	1	To investigate whether *BRAF* mutant CRC is further defined by metastatic spread and evaluation of the impact of this mutation on prognosis.
Yaeger et al. [49]	2014	USA	CC	2009	2012	-	515	1	To determine the clinicopathologic characteristics, *PIK3CA* mutation frequency, and outcomes after metastasectomy in patients with *BRAF*-mutant mCRC.
Yang et al. [50]	2021	China	CH	01/2015	03/2020	Chemo- or radiotherapy before NGS and no follow-up information.	582	1	To evaluate the frequency and phenotypic characteristics of mCRC with somatic RET mutation.
Yokota et al. [40]	2011	Japan	CH	2002	2010	-	229	2	To investigate the clinicopathological features and prognostic impact of *KRAS*/*BRAF* mutation in advanced and recurrent CRC patients.
Zihui Yong et al. [51]	2018	Singapore	CH	01/2010	12/2014	Appendiceal tumors, other stages than stage 4 and patients without metastases.	363	1	To describe the metastatic pattern of advanced CRC by assessing the interaction between the *KRAS* mutational status and the location of primary tumors.

CC, case control study; CH, cohort study; CRC, colorectal cancer; CRS, cytoreductive surgery; FAP, familiarly adenomatous polyposis; HNPCC, hereditary non-polyposis colon cancer; HIPEC, hyperthermic intraperitoneal chemotherapy; MCH, matched cohort study; mCRC, metastatic colorectal cancer; PM, peritoneal metastases.

**Table 2 cancers-15-00549-t002:** Characteristics of patients and tissue samples in included studies.

Reference	Patients’ Specification	No. of Patients for Analysis		Tissue Samples	Tissue Collection
		Without PM	With PM		
Astrosini et al. [20]	Nonpretreated CRC patients.	54	9	Primary tumor tissue	Retrospect, archives of pathology department
Atreya et al. [43]	Cohort of patients with *BRAF* mutant mCRC was matched 1:2 to patients with *BRAF* WT mCRC.	75	45	Not specified	Not performed, *BRAF* status was already known
Bruzzi et al. [21]	Patients with signed informed consent for biological sample collection from the PETACC-8 trial (stage III CRC patients) were included.	1446	38	FFPE tumor tissues	Retrospect, archives of pathology department
Cheng et al. [22]	Patients who underwent surgery for CRC. Stage IV patients are included for genetic analysis.	260	76	Frozen tumor tissue in liquid nitrogen	Retrospect, hospital biobank
Christensen et al. [44]	Patients with mCRC who had received multiple treatment line (irinotecaban and cetiuximab).	405	43	FFPE tumor tissue blocks from primary tumors	Not performed, *BRAF*, RAS and *PIK3CA* status was already known
He et al. [23]	Therapy-naïve synchronous mCRC patients at first diagnosis.	168	26	FFPE tumor tissue blocks	Retrospect, archives of pathology department
Heublein et al. [24]	Patients underwent surgical resection and were divided into three groups according to metastases location.	10	13	FFPE tissue from primary tumors	Retrospect, archives of pathology department
Jacob et al. [25]	Patients undergoing surgery. Four groups; patients without metastases, with LM, with PM and with LM and PM.	12	6	FFPE tissue from primary tumors	Retrospect, national database and biobank
Jacob et al. [26]	CRC patients surgically treated and divided in three groups: patients without metastases, with LM and with PM.	12	6	FFPE tissue from primary tumors	Retrospect, archives of pathology department
Kawazoe et al. [27]	CRC patients with histologically confirmed adenocarcinoma and presence of unresectable metastatic disease.	212	52	FFPE cancer specimens (239 primary tumors and 25 metastases)	Retrospect, archives of pathology department
Lan et al. [28]	Patients with stages I–IV CRC who underwent surgery.	1388	104	Surgery tissue samples	Retrospect, hospital biobank
Lan et al. [29]	Patients with stage IV CRC who underwent surgery.	63	32	Surgery tissue samples	Retrospect, hospital biobank
Lee et al. [30]	Small obstructive colorectal cancer group compared to large non-obstructing colorectal cancer group (contrast group).	5	10	FFPE surgical specimens	Retrospect, archives of pathology department
Nagahara et al. [31]	Patients identified as having primary CRC based on the clinicopathologic criteria described by the Japanese Society for Cancer of the Colon and Rectum.	107	6	Frozen tissue specimens in liquid nitrogen from colorectal tumor tissue and paired healthy tissue at least 10 cm distal from primary tumor	Retrospect, archives of pathology department
Prasanna et al. [45]	Patients with proven mCRC who were registered in the included databases.	4755	1212	Unknown	Not performed, *BRAF* and *KRAS*/RAS status was already known
Roberto et al. [46]	Patients with right mCRC with known *BRAF* mutation status.	154	53	Unknown	Not performed, *BRAF* status was already known
Sakuraba et al. [32]	Patients undergoing surgery for CRC.	33	5	Frozen tumor specimens and corresponding normal tissues	Retrospect, archives of pathology department
Sasaki et al. [33]	Patients with mCRC treated with systemic chemotherapy, combined with or without bevacizumab, cetuximab or panitumumab.	409	117	FFPE tumor samples	Retrospect, archives of pathology department
Sayagués et al. [34]	Caucasian patients diagnosed with CRC who underwent surgical resection of primary tumor tissues.	80	7	Freshly frozen primary tumor tissues	Retrospect, archives of pathology department
Schirripa et al. [47]	Patients with presence of a *KRAS* mutation with the focus of the specific variant.	391	108	FFPE tissue from primary tumors and/or paired metastases	Not performed, *KRAS* status was already known
Shelygin et al. [35]	Patients undergoing surgery for colorectal cancer.	38	20	Tissue samples from primary tumor, peritoneal metastases, and healthy tissue	Retrospect, archives of pathology department
Shirahata et al. [42]	CRC patients who underwent surgery.	47	5	Frozen colorectal cancer tissue and corresponding normal tissues	Were collected at surgical resection and stored at the pathology department
Shirahata et al. [41]	CRC patients who underwent surgery.	43	5	Frozen primary tumor specimens and corresponding normal tissue	Were collected at surgical resection and stored at the pathology department
Sjo et al. [36]	CRC patients with PM.	57	148 *	FFPE tumor samples	Retrospect, archives of pathology department
Smith et al. [37]	Patients with measurable metastatic of locally advanced colorectal adenocarcinoma and unresectable disease.	1667	283	FFPE tumor samples (adenocarcinomas)	Retrospect, archives of pathology department
Takahashi et al. [38]	Patients with CRC who underwent surgical treatment.	174	6	Frozen resected tumor samples in liquid nitrogen	Retrospect, archives of pathology department
Taniguchi et al. [39]	Patients with CRC who received any treatment at one of the 15 study hospitals that participated, and who had RAS WT tumors.	62	281	FFPE tumor samples	Retrospect, archives of pathology department
Tran et al. [48]	Patients with mCRC with known *BRAF* mutation status from two institutional databases.	385	139	Tumor tissue	Not performed, *BRAF* status was already known
Yaeger et al. [49]	Patients with mCRC with available tumor sequencing.	431	84	FFPE primary tumor samples or metastatic tissue	Not performed, *BRAF* status was already known
Yang et al. [50]	Cohort of patients with mCRC.	412	170	Unknown	Not performed, RET status was already known
Yokota et al. [40]	Cohort of patients with CRC.	175	54	Frozen or FFPE tissues	Retrospect, archives of pathology department
Zihui Yong et al. [51]	Stage 4 CRC patients with metastases to the liver, lung, and/or peritoneum.	266	89	FFPE surgical specimens	Not performed, *KRAS* status was already known

CRC, colorectal cancer; FFPE, formalin-fixed paraffin embedded; LM, liver metastases; mCRC, metastatic colorectal cancer; PM, peritoneal metastases; * Use of a control group with non-PM from a previous published study.

**Table 3 cancers-15-00549-t003:** Overview of genetic analysis and outcomes.

Reference	Level of Testing	Name Genes, Molecules or Panel Investigated	Type of Analysis Performed	Gene or Molecule Name and Mutation or Expression Status (n)	No. of Patients with PM (N) and Outcomes (n)		No. of Patients without PM (N) and Outcomes (n)	MMR Status (MSI/MSS)	Findings as Reported by Authors in Studies
Astrosini et al. [20]	RNA	*REG1A*	RT-PCR		*N* = 9		*N* = 54	N/A	*REG1A* expression levels highly correlated with formation of PM (median relative amount of 10.36 vs. 0.94, *p* = 0.0039 ^a^).
				*REG1A* expression	-		-		
Atreya et al. [43]	DNA	*BRAF*	-		*N* = 45		*N* = 75	Total: 10/68 PM: 2/23	No significant differences in metastatic sites were observed, although PM were more common in *BRAF* mutant patients (*p* = 0.045 ^†,b^).
				*BRAF* mutant (40)	20		20		
				*BRAF* wild-type (80)	25		55		
Bruzzi et al. [21]	DNA	*BRAF*, *RAS* (*KRAS* and *NRAS*)	RT-PCR		*N* = 38		*N* = 1446	Only MSS included.	There is a trend for a higher rate of PM in *BRAF*V600E mutant compared to *RAS* mutant and wild-type patients (12.2% vs. 7.44% vs. 9.96% respectively, *p* > 0.05 ^c,d^).
				*BRAF* V600E mutant (127)	15		112		
				RAS mutant (748)	56		692		
				Double wild-type (609)	61		548		
Cheng et al. [22]	DNA	*BRAF*	PCR or SNP genotyping assay		*N* = 76		*N* = 260	N/A	Stage IV CRC patients with a *BRAF*V600E mutation had a higher frequency of PM (41.7% vs. 21.2%, *p* = 0.04 ^d^).
				*BRAF* V600E mutant (312)	66		246		
				*BRAF* wild-type (24)	10		14		
Heublein et al. [24]	miRNA	miRNAs	RT-PCR		*N* = 10		*N* = 13	N/A	A set of 31 miRNAs was significantly upregulated in the PM group, while 10 miRNAs were repressed as compared to LM. A set of 2 miRNAs was significantly upregulated in the PM group, while 25 were repressed as compared to M0. hsa-miR-31-5p was significantly overexpressed in PM patients (*p* = 0.002 ^b,d^).
				hsa-miR-215-5p	Induced 17-fold		Compared to LM		
				hsa-miR-31-3p	Induced 8.9-fold		Compared to LM		
				hsa- miR-31-5p	Induced 5.4-fold		Compared to LM		
				hsa-miR-483-5p	Repressed 0.04-fold		Compared to LM		
				hsa-miR-1226-5p	Repressed 0.29-fold		Compared to LM		
				hsa-miR- 296-5p	Repressed 0.32-fold		Compared to LM		
				hsa-miR-215-5p	Induced 3.6-fold		Compared to M0		
				hsa-miR-148a-3p	Induced 2.8-fold		Compared to M0		
Jacob et al. [25]	mRNA	PanCancer Progression Panel	NanoString analysis		*N* = 6		*N* = 12	N/A	In PM patients, 18 genes demonstrated significant different expression rates (*p* < 0.05 ^b,d^).
				Not specified	-		-		
Jacob et al. [26]	RNA	PanCancer Progression Panel	NanoString analysis		*N* = 6		*N* = 12	N/A	The analysis between patients with PM and M0 did not show a significant down- or upregulation of distinct gene sets.
				Not described	-		-		
Kawazoe et al. [27]	DNA	*KRAS*, *NRAS*, *BRAF* and *PIK3CA*	PCR		*N* = 52		*N* = 212	N/A	*BRAF* mutant tumors were more likely to have PM in comparison with *BRAF* wild-type tumors (50.0% vs. 18.0%, *p* = 0.009 ^c^). No significant differences for PM according to *RAS* mutation (*p* = 0.64 ^d^).
				*BRAF* mutant (14)	7		7		
				*RAS* pathway mutant (21)	4		17		
				*KRAS* exon 2 mutant (90)	15		75		
				*PIK3CA*	N/A		N/A		
Lan et al. [28]	DNA	RAS pathway (*KRAS*, *NRAS*, *HRAS*, *BRAF*) *PI3K* pathway	PCR		*N* = 104		*N* = 1388	Total: 154/1492	PM was significantly higher in RAS pathway mutated patients compared to wild-type tumors (*p* = 0.009 ^d^). Tumors with *KRAS* mutation had a trend toward a higher proportion of PM (*p* = 0.061 ^d^). There was no association between PM and the presence of a *PI3K* pathway mutation (*p* = 0.408 ^d^).
				*PI3K* pathway mutant (213)	12		201	36/177	
				*RAS* pathway mutant (706)	62		644	91/615	
				*BRAF* mutant (70)	8		62	N/A	
				*KRAS* mutant (602)	51		551		
				*NRAS* mutant (49)	5		44		
				HRAS mutant (21)	4		17		
Lan et al. [29]	DNA	*TP53*, APC, *KRAS*, *FAT4*, *ARID1A*, *FBXW7*, *SMAD4*, *PIK3CA*, *NRAS* and *BRAF*	NGS		*N* = 32		*N* = 63	N/A	For patients with PM, the frequency of genetic mutations was the highest in *TP53*. The authors conducted analysis to compare left- and right-sided CRC with mutation status but not between the non-PM and PM group.
				*TP53* mutant (59)	19		40		
				*KRAS* mutant (35)	7		28		
				APC mutant (45)	17		28		
Lee et al. [30]	DNA	Life Technologies Ion AmpliSeq Comprehensive Cancer Panel	NGS		*N* = 10		*N* = 5	N/A	*ARID1A*, *PKHD1*, *UBR5*, *PAX5*, *TP53*, *ASXL1* and *AR* were detected more frequently in the SOC group with PM (*p* values of 0.002, 0.019, 0.002, <0.001, 0.007, 0.047, 0.019 respectively ^d^). *TNFRSF14*, *VHL*, *MTRR*, *MLLT10*, *BIRC2*, *EP400*, *IRS2*, *PER1*, *TCF3* and *CYP2D6* were detected more frequently in the LNOC group without PM (*p* values of 0.019, 0.004, 0.047, 0.022, <0.001, 0.022, <0.001, 0.022, 0.026, 0.004 respectively ^d^).
				*ARID1A* mutant	9		0		
				*PKHD1* mutant	7		0		
				*UBR5* mutant	9		0		
				*PAX5* mutant	10		0		
				*TP53* mutant	8		0		
				*ASXL1* mutant	8		1		
				*AR* mutant	7		0		
				*TNFRSF14* mutant	5		3		
				*VHL* mutant	4		0		
				*MTRR* mutant	4		2		
				*MLLT10* mutant	3		0		
				*BIRC2* mutant	5		0		
				*EP400* mutant	3		0		
				*IRS2* mutant	5		0		
				*PER1* mutant	3		0		
				*TCF3* mutant	5		3		
				*CYP2D6* mutant	4		0		
Nagahara et al. [31]	RNA	*Kif18A*	RT-PCR		*N* = 6		*N* = 107	N/A	*Kif18A* overexpression in CRC significantly correlated with PM (*p* = 0.02 ^d^).
				*Kif18A* low expression	0		38		
				*Kif18A* high expression	6		69		
Prasanna et al. [45]	-	*BRAF*, RAS	-		*N* = unknown		*N* = unknown	PM: 29/239 M0: 77/940	*BRAF*-mutated colorectal cancer showed higher incidence of PM with a relative risk of 1.8 (*p* < 0.001 ^e^). *KRAS*-mutated patients showed no higher incidence of PM with a relative risk of 0.95 (*p* = 0.63 ^e^).
				*RAS* mutant (965)	199		766		
				*RAS* wild-type (1271)	274		997		
				*BRAF* mutant (143)	51		92		
				*BRAF* wild-type (1058)	208		850		
Roberto et al. [46]	-	*BRAF*	-		*N* = 53		*N* = 154	Total: 19/66	*BRAF* mutant right colorectal cancer was significantly more likely to occur with peritoneal metastases (38.1% vs. 22.4%, *p* = 0.003 ^d^).
				*BRAF* mutant (42)	16		26		
Sakuraba et al. [32]	RNA	*Tip60*	RT-PCR		*N* = 5		*N* = 33	N/A	The authors found that *Tip60* downregulation (compared to healthy tissue expression) showed significant correlation with PM (*p* = 0.0053 ^d^).
				Downregulation of *Tip60* expression	3		2		
Sasaki et al. [33]	DNA	*BRAF*, *KRAS*, *PIK3CA*	PCR		*N* = 117		*N* = 409	N/A	The PM group had a significantly higher incidence of the *BRAF* V600E mutation than the non-PM group (27.7% vs. 7.3%, *p* < 0.01 ^d^). In contrast, no differences were observed between the two groups in *KRAS* and *PIK3CA* mutations (*p* 0.42 ^d^ and 0.76, ^d^ respectively).
				*KRAS* wild-type	54		163		
				*KRAS* mutation	46		115		
				*BRAF* wild-type	34		115		
				*BRAF* mutation	13		9		
				*PIK3CA* wild-type	53		181		
				*PIK3CA* mutation	5		*20*		
Sayagués et al. [34]	DNA	*KRAS*/*NRAS*, *BRAF* and *TP53*	NGS		*N* = 7		*N* = 80	Total: 6/48	*BRAF*-mutated CRC tumors were significantly associated with PM (*p* = 0.006 ^d^).
				*KRAS* mutant (24)	1		23	0/16	
				*NRAS* mutant (1)	0		1	0/0	
				*BRAF* mutant (6)	3		3	3/2	
				*TP53* mutant (29)	2		*27*	1/21	
Schirripa et al. [47]	-	*KRAS*	Sanger sequencing, Sequenom MassArray		*N* = 108		*N* = 391	N/A	Compared to other *KRAS*-mutated cases, *KRAS G12C* mutations had a lower frequency in PM patients (13.5% vs. 25%, *p* = 0.008 ^d^).
				*KRAS* mutant (694)	90		276		
				*KRAS* G12C mutant (145)	18		115		
Shelygin et al. [35]	DNA and RNA	*KRAS*, *BRAF* and *EMT* *	RT-PCR		*N* = 20		*N* = 38	PM: 2/18 M0: 6/32	Mutations in *KRAS* and *BRAF* with PM was 70%, compared to 42.1% in M0 patients (*p* = 0.04 ^d^). The frequency of wild types in both genes was 57.9% in CRC without PM compared to 30% with PM (*p* = 0.04 ^d^). No differences were observed between the two groups in *KRAS* and *BRAF* mutations solely.
				*KRAS* mutant	11		15		
				*BRAF* V600E	3		1		
				*KRAS*/*BRAF* wild-type	6		22		
				*KRAS*/*BRAF* mutant	14		16		
				*EMT* + (13)	7		6		
				*EMT* − (45)	13		32		
Shirahata et al. [42]	RNA	*MACC1* expression	RT-PCR		*N* = 5		*N* = 47	N/A	*MACC1* expression showed significant correlation with PM compared to the group without PM (5.75 ± 4.58 vs. 2.57 ± 3.09, *p* = 0.042 ^b^).
				*MACC1* expression	-		-		
Shirahata et al. [41]	DNA	Vimentin methylation	qMSP		*N* = 5		*N* = 43	N/A	A trend was shown toward preferentially developing PM with Vimentin methylation (*p* = 0.080 ^d^).
				Vimentin +	5		26		
				Vimentin -	0		17		
Sjo et al. [36]	DNA	*TP53*	PCR		*N* = 49		*N* = 148	N/A	Univariate analyses demonstrated that PM was significantly associated with mutations in the *TP53* gene (*p* = 0.05 ^f^). Multivariate analyses confirmed the previous finding (OR = 2.4; 95%CI = 1.2–4.8, *p* = 0.013 ^g^).
				*TP53* mutant	-		-		
				*TP53* wild-type	-		-		
Smith et al. [37]	DNA	*KRAS*, *NRAS*, *BRAF*, and *PIK3CA*	PS and Sequenom		*N* = 283		*N* = 1667	Co-occurred with *BRAF* wild-type tumors	*BRAF* mutations were more common in patients with PM-only compared to LM-only (22.2% vs. 6.7%, *p* = 0.00092 ^d^), although this association with PM did not withstand correction for multiple testing. *BRAF* mutations were significantly associated with PM (*p* = 0.018), which did not remain significant after Bonferroni correction (*p* = 0.36). For *KRAS*, *NRAS*, and *PIK3CA*, there was no association found for PM.
				*KRAS* mutant	131 (/282)		693 (/1667)		
				*BRAF* mutant	36 (/283)		193 (/1663)		
				*NRAS* mutant	7 (/283)		62 (/1656)		
				*PIK3CA* mutant	40 (/280)		293 (/1627)		
Takahashi et al. [38]	RNA	*NEK2*	RT-PCR		*N* = 6		*N* = 174	N/A	The high *NEK2* expression group showed greater PM then the low *NEK2* mRNA expression group (*p* = 0.004 ^b,d^).
				*NEK2* low (90)	0		90		
				*NEK2* high (90)	6		84		
Taniguchi et al. [39]	DNA	*RAS*, *NRAS*, *BRAF* and *PIK3CA*	PCR		*N* = 62		*N* = 281	N/A	The frequencies of RAS/*BRAF*V600E wild-type over either *BRAF* or *PIK3CA* mutations were higher for PM (35% vs. 15%, *p* = 0.003 ^d^).
				RAS and *BRAF* wild-type (291)	44		247		
				RAS wild-type + *BRAF* or *PIK3CA* mutation (52)	18		34		
Tran et al. [48]	DNA	*BRAF*	RT-PCR		*N* = 139		*N* = 385	Total: 40/310	*BRAF* mutant tumors had significantly higher rates of PM (46% vs. 24%, *p* = 0.001 ^d^).
				*BRAF* mutant (57)	26		31	12/30	
Yaeger et al. [49]	DNA	*BRAF*	Sanger sequencing, Sequenom MassArray		*N* = 84		*N* = 431	N/A	PM was significantly more common at the time of diagnosis of metastatic disease in the *BRAF*-mutant cases (26% vs. 14%, *p* < 0.01 ^d^).
				*BRAF* mutant (92)	24		68		
Yang et al. [50]	DNA	*RET*	NGS		*N* = 170		*N* = 412	Total: 24/558	The presence of *RET* mutations was associated with PM compared to wild-type tumors (56.2% vs. 28.4%, *p* = 0.024 ^d^).
				*RET* mutant (16)	9		7	6/10	
Yokota et al. [40]	DNA	*KRAS* and *BRAF*	PCR		*N* = 54		*N* = 175	N/A	60.0% of CRCs with *BRAF* mutation develops PM compared with 15% of CRCs with other subtypes (*p* = 0.0062 ^b,d^).
				*KRAS*/*BRAF* wild-type (135)	30		105		
				*KRAS*G12X mutant (53)	11		42		
				*KRAS*G13X mutant (26)	4		22		
				*BRAF*V600E mutant (15)	9		6		
Zihui Yong et al. [51]	DNA	*KRAS*	PCR		*N* = 89		*N* = 266	N/A	After stratification, PM was associated with mutant *KRAS* tumors (26.6% vs. 15.1%, *p* = 0.02 ^d^).
				*KRAS* mutant (126)	37		89		
**Studies with synchronous and metachronous peritoneal metastases population**									
**Reference**	**Level of Testing**	**Genes or Panel** **Investigated**	**Type of Analysis**	**Gene or Molecule Name and Mutation or Expression Status (n)**	**No. of Patients** **with PM (N) with Outcomes (n)**		**No. of Patients** **without PM (N) with Corresponding Outcomes (n)**	**MMR** **Status (MSI/MSS)**	**Conclusions** **Findings as Reported by** **Authors in Studies**
					**Synch.**	**Metach.**			
Christensen et al. [44]	DNA	RAS (*KRAS* and *NRAS*), *BRAF* and *PIK3CA*	NGS or Mutation kit and PS		*N* = 43	*N* = 33	*N* = 372	N/A	*PIK3CA* mutations were significantly associated with absence of PM (OR = 0.10; 95%CI = 0.01–0.79, *p* = 0.028 ^g^) and with a decreased hazard of developing PM (HR = 0.31; 95%CI = 0.11–0.86, *p* = 0.024 ^g^). The hazard ratio of developing PM and having *BRAF* mutations were not associated with PM (OR = 2.07; 95%CI = 0.60-6.19, *p* = 0.192 ^g^ and (HR = 1.82; 95%CI = 0.81–4.08, *p* = 0.146 ^g^).).
				RAS mutant (206)	21	16	169		
				*BRAF* V600E mutant (30)	7	3	20		
				*PIK3CA* mutant (61)	1	3	57		
He et al. [23]	DNA	*KRAS*, *BRAF*, *NRAS*	NGS		*N* = 26	*N* = 0	*N* = 174	N/A	Mutant *KRAS* tumors had a significant relevance with PM (*p* = 0.017 ^d^). *KRAS* codon 12 mutation was more likely to present with PM (*p* = 0.014 ^d^). Patients with PM had the tendency to carry mutant *KRAS* G12D (*p* = 0.052 ^d^). Tumors with mutated *BRAF* were more likely to develop PM (*p* = 0.052 ^d^).
				Any mutation (108)	20	-	88		
				*KRAS* mutant (77)	15	-	62		
				*NRAS* mutant (8)	0	-	8		
				*BRAF* mutant (23)	5	-	18		
				All wild-type (86)	6	-	80		

*EMT*, epithelial–mesenchymal transition; FFPE, formalin fixated paraffin embedded; LNOC, large non-obstructing colorectal cancer; LM, liver metastases; MS, microsatellite; MMR, mismatch repair; MSS, microsatellite stable; MSI, microsatellite instable; N/A, non-applicable; NGS, next-generation sequencing; PCR, polymerase chain reaction; PM, peritoneal metastases; PS, pyrosequencing; qMSP, quantitative methylation-specific polymerase chain reaction; RT, quantitative real time; SOC, small obstructing colorectal cancer; SNP, single nucleotide polymorphism. * *EMT*, epithelial–mesenchymal transition (*EMT*) = *ZEB* 1, *ZEB 2*, *SNAI1* and *VIM* overexpression and *CDH1* downregulation. ^†^ No longer met the criteria for statistical significance. ^a^ Kruskal–Wallis test; ^b^ Student’s t-test; ^c^ Mann–Whitney U test; ^d^ Chi-square test or Fisher’s exact test; ^e^ Mantel–Haenszel chi-squared test; ^f^ Univariate analysis; ^g^ Multivariate analysis.

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
