# Peer review of "DNA and RNA Alterations Associated with Colorectal Peritoneal Metastases: A Systematic Review"

_cancers, 2023, doi:10.3390/cancers15020549_

Round 1
Reviewer 1 Report
Thanks to the authors for their manuscript.
Although comprehensive, the manuscript is quite long. Tables 1, 2 and 3 are represent a large number of pages in the manuscript. Is there a way to merge tables?
Possible options: Two tables rather than 3? Perhaps less important data elements could be put an Appendix? E.g. Country of origin, reasons for patient exclusion, some other less important elements for most readers but still available for those doing a more detailed read. The amount of text in the tables may be too extensive. Is there a way to summarize individual conclusions of publications?
Beyond this main issue, the manuscript is well-written.
Author Response
Dear reviewers and editor,
First of all, my best wishes for the New Year. I want to thank you for your review of “DNA and RNA alterations associated with colorectal peritoneal metastases: a systematic review.” (ID cancers-2124667), submitted to Cancers (ISSN 2072-6694).
I greatly appreciate the time and commitment that goes into each review. Cancers would not be the quality publisher that it is without your effort, wherefore thank you. We sincerely appreciate your feedback and suggestions, which helped us in improving the quality of the manuscript. After careful consideration of all your comments and those of the other reviewer, we made a revision of the manuscript.
Regarding the comments on the tables, Cancers will work on the layout (e.g., landscape orientation) to make them easier to read.
We will upload a new document, entitled “cancers-2124667”, shows all changes in the original manuscript using ‘track changes’. The document attached to this reply, entitled "letter to the reviewer", contains all reactions to your comments.
Let me know if you have any additional questions or comments.
Thanks again.
Best regards,
Danique Heuvelings
On behalf of all co-authors

Reviewer 2 Report
Based on previously published evidence, this systematic review aimed at providing novel biomarkers capable of predicting the emergence of peritoneal metastases in CRC patients, as well as potential biomarkers assisting in prognostic evaluation or treatment response prediction in patients with synchronous peritoneal metastasis. The main findings of this systematic review are not entirely outstanding and, as the authors have already explained in their study, more in-depth and homogeneous primary research is needed in order to perform a meta-analysis and obtain more reliable results. However, the topic of the study is very interesting, being also well-organized and the methodology employed is appropriate. In addition, this study sets the basis for larger investigations related to this subject. Overall, there are some issues that should be addressed. Find below the minor comments to each section:
Comments to Abstract:
1. The abstract should be changed in order to be structured but without headings. Moreover, information regarding conflicts of interest and funding should not appear in the abstract.
Comments to Keywords:
2. Keywords should be rearranged in alphabetical order.
Comments to Introduction:
3. The abbreviation PIK3CA should be placed in parentheses instead of its definition.
Comments to Methods:
4. PRISMA 2020 statement (Page MJ, McKenzie JE, Bossuyt PM, Boutron I, Hoffmann TC, Mulrow CD, et al. The PRISMA 2020 statement: an updated guideline for reporting systematic reviews. BMJ 2021;372:n71. doi:10.1136/bmj.n71) should be used instead of PRISMA 2009 statement. The authors have to update PRISMA guidelines in their manuscript, including both the new PRISMA 2020 flow diagram (Figure 1 updating is required) and the PRISMA 2020 checklist.
5. Taking into account that 10 months elapsed from the first literature search performed to the last one (November 3rd, 2022), were the studies published during that period also included in the current systematic review? Curiously, there is not any investigation published in 2022 included in the present systematic review.
Comments to Results:
6. In 4.2. subsection, it should be specified that this systematic review has included both cohort and case control studies.
7. In 4.2. subsection, the authors have mentioned that tissue samples had retrospectively been retrieved in 21 studies, but they have provided the references of 23 studies: from 20 to 42. Likewise, the authors have reported that eight studies had used fresh frozen tumor samples, but they have only provided the references of seven articles: 22, 31, 32, 34, 36, 37, 40). These issues should be corrected.
8. The authors have stated that 24 tissue samples had been FFPE. Nonetheless, instead of 24 samples, it seems that all tissue samples from 24 out of the 32 investigations included in the systematic review used this method. This issue needs to be clearly detailed.
9. All tables in this systematic review should be displayed in a more suitable layout since the information in the current table format is very difficult to follow and read. Tables could be placed horizontally instead of vertically.
10. The abbreviation pCH in the footnote of Table 1 appears to be unnecessary.
11. In 4.3. subsection, it should be mentioned that the lower risk of bias was not only found in the statistical domain but also in the outcome measurement domain.
12. In Figure 2, (A) and (B) should be placed above the corresponding panels.
Comments to Results of individual studies:
13. Considering that the title of the previous section was “Results” and also included data from individual studies, titles of sections 4 and 5 should be more specific and better summarize the information contained in these sections.
14. All gene abbreviations that appear throughout the manuscript should be defined in the main text the first time they appear. Furthermore, one additional column containing the full gene names could be added to Table 4.
15. There are some abbreviations in Table 3 that are not defined in the corresponding table footnote (e.g. N/A, SNP, PTEN, etc.). The authors should review the abbreviations in this Table and add the missing definitions.
16. All abbreviations and its definitions should be rearranged in alphabetical order in all table footnotes.
17. In Table 3 footnote, MRR abbreviation seems to be misspelled. MRR should be replaced with MMR.
18. In 5.4. subsection, the references of those 17 articles where BRAF mutations were analyzed should be provided in the main text.
19. In 5.9. subsection, results from the study by Lee at al. (reference 30) should be added, as that study also analyzed a broader cancer panel.
20. The title of the second column of Table 4 should be replaced with a more suitable one, since it refers to the location of the genetic alteration rather than the gene location.
Author Response

(The authors gave the same response as above.)
